# How Does the Method Used to Measure the VE/VCO_2_ Slope Affect Its Value? A Cross-Sectional and Retrospective Cohort Study

**DOI:** 10.3390/healthcare11091292

**Published:** 2023-04-30

**Authors:** Martin Chaumont, Kevin Forton, Alexis Gillet, Daryl Tcheutchoua Nzokou, Michel Lamotte

**Affiliations:** Department of Cardiology, Erasme Hospital, 1070 Brussels, Belgium

**Keywords:** CPET, VE/VCO_2_ slope, VO_2_peak

## Abstract

Cardiopulmonary exercise testing (CPET) was limited to peak oxygen consumption analysis (VO_2_peak), and now the ventilation/carbon dioxide production (VE/VCO_2_) slope is recognized as having independent prognostic value. Unlike VO_2_peak, the VE/VCO_2_ slope does not require maximal effort, making it more feasible. There is no consensus on how to measure the VE/VCO_2_ slope; therefore, we assessed whether different methods affect its value. This is a retrospective study assessing sociodemographic data, left ventricular ejection fraction, CPET parameters, and indications of patients referred for CPET. The VE/VCO_2_ slope was measured to the first ventilatory threshold (VT1-slope), secondary threshold (VT2-slope), and included all test data (full-slope). Of the 697 CPETs analyzed, 308 reached VT2. All VE/VCO_2_ slopes increased with age, regardless of test indications. In patients not reaching VT2, the VT1-slope was 32 vs. 36 (*p* < 0.001) for the full-slope; in those surpassing VT2, the VT1-slope was 29 vs. 33 (*p* < 0.001) for the VT2-slope and 37 (all *p* < 0.001) for the full-slope. The mean difference between the submaximal and full-slopes was ±4 units, sufficient to reclassify patients from low to high risk for heart failure or pulmonary hypertension. We conclude that the method used for determining the VE/VCO_2_ slope greatly influences the result, the significant variations limiting its prognostic value. The calculation method must be standardized to improve its prognostic value.

## 1. Introduction

The maximal cardiopulmonary exercise test (CPET) consists of exercise with a gradually increasing intensity until exhaustion or the appearance of limiting symptoms [1]. The main parameters measured are ventilation (VE), oxygen consumption (VO_2_), and carbon dioxide production (VCO_2_). The CPET is extremely useful for the evaluation of cardiovascular, respiratory, muscular, and metabolic systems during exercise and is considered the gold standard for evaluating cardiopulmonary function [1,2]. It is used, notably, for the prognosis or follow-up assessment of patients with cardiac or respiratory diseases in the pre-operative period, or the prescription of physical exercises for athletes or patients [3,4]. 

Early prognostic studies about the CPET were limited to the analysis of maximal aerobic power by a measure of the peak oxygen consumption (VO_2_peak). This parameter is used to characterize the functional severity of heart failure (HF) and as a criterion for listing heart transplantations and preoperative risk stratification [5,6]. A drawback to the use of the VO_2_peak is that factors unrelated to either cardiac or pulmonary pathology can falsely reduce its value. For example, if a patient is exercise-intolerant or lacks motivation to reach the maximal effort, the VO_2_peak interpretation could be limited [6]. In addition, the VO_2_peak resulting from a submaximal test produces a false, poorly predictive score compared with that from a maximal test [7]. Due to these potential difficulties in estimating the VO_2_peak, along with confounding factors such as age, sex or obesity, experts have sought other primary CPET factors to improve its predictive value [5,6,7,8]. The relationship between VE and VCO_2_, expressed as the VE/VCO_2_ slope, has been identified and documented [9]. A normal ventilatory response of the VE/VCO_2_ slope should be under 30.0 [2]. Patients with cardiopulmonary disease often present an abnormal ventilation response, characterized by an increase in the VE/VCO_2_ slope; for any given VCO_2_ level, VE is greater than normal.

The VE/VCO_2_ slope, VO_2_peak and presence of exercise oscillatory ventilation (EOV) are considered primary CPET variables and are strong predictors of adverse events in heart failure (Class 1A) [2]. In addition, the VE/VCO_2_ slope is responsive to pharmacological, surgical, and exercise interventions in HF patients and can be used to assess therapeutic efficacy (Class 1A) [2]. In patients with chronic HF, a Kaplan–Meier analysis revealed a 1-year cardiac-related mortality of 75% in patients with a VE/VCO_2_ slope >35.6 and 25% in those with a VE/VCO_2_ slope <35.6; 1-year cardiac-related hospitalization was 77% in patients with a VE/VCO_2_ slope >32.5 and 23% in those with a VE/VCO_2_ slope <32.5 [6].

From a physiological point of view, the relationship of VE and VCO_2_ is conserved during submaximal exercise, characterizing the ventilatory exercise response of the patient. The VE/VCO_2_ slope from rest to VT2 is linear; therefore, an accurate VE/VCO_2_ slope should be obtained even if the exercise test is submaximal, making it more feasible [10].

To date, there are several methods of calculating the VE/VCO_2_ slope, varying from one study to another, and expert recommendations are not that clear. Some authors calculate the slope only using the data from the start of exercise to the aerobic threshold (i.e., first ventilatory threshold (VT1)) or the ventilatory compensation point (i.e., secondary ventilatory threshold (VT2)), while others use all the data from the start to the maximum effort (full-slope) [10]. Thus, during a single CPET, several VE/VCO_2_ slope measurements can be obtained depending on the method used. This measurement heterogeneity certainly limits the clinical relevance and prognostic value of the VE/VCO_2_ slope when the test is maximal. This work aims to evaluate and compare the different methods of measuring the VE/VCO_2_ slope, considering values from rest up to VT1 (VT1-slope), to VT2 (VT2-slope), and including all the data from the entire test (full-slope). We formulated the primary hypothesis that the gradient of the VE/VCO_2_ slope is greater for all time intervals (i.e., up to VT1, VT2, and over the entire test). In addition, we hypothesized that the VE/VCO_2_ slope is influenced by age and cardiopulmonary disease.

## 2. Materials and Methods

### 2.1. Study Setting

We carried out a retrospective cross-sectional study at the cardiology unit of the Erasme Hospital, Brussels, Belgium, between May and December 2019. Data from 697 successive CPETs carried out on a cycloergometer were included in our database. We used a Strobe checklist for retrospective study.

### 2.2. Participants

All patients referred for CPET as part of their medical work-up were included in the study: Cardiac patients were referred for HF or pulmonary hypertension follow-up or the beginning of cardiac rehabilitation after ischemic or valvular surgery; pulmonary populations were addressed for pulmonary fibrosis, chronic obstructive pulmonary disease (COPD) follow up or pre-operative thoracic surgery; the dyspnea group was composed of patients addressed for exercise testing after a normal cardiac and pulmonary screening; and the last group (other) was composed of oncology, obesity (but normal cardiac screening) or rheumatoid arthritis patients. Patients excluded were those with Gold III and IV (COPD), those who did not achieve VT1, those who performed a submaximal effort (i.e., peak respiratory exchange ratio (RERpeak) < 1.10), and those whose VT1 and VT2 could not be accurately determined.

### 2.3. Procedure

CPET was performed on a cycloergometer (Ergoselect II 1200; Ergoline, Bitz, Germany) with a step-by-step increase in load mode. CPET consisted of a three-minute warm-up followed by increments of 10–30 W per minute in a standing position. The increments were chosen according to the individual’s estimated capacity, such that the duration of the effort was between 8 and 12 min. Patients were required to maintain a cycling cadence of 60 rpm. VO_2_ (oxygen uptake), VCO_2_ (carbon dioxide output), and ventilation (VE) were collected breath by breath via a facial mask and analyzed every 8 s using a metabolic system (Exp’Air^®^, Medisoft, Dinant, Belgium). The flow in the pitot tube was calibrated with a 3 L syringe, which was annually calibrated, and verified by the manufacturer. The O_2_ and CO_2_ fractions were measured by laser cells calibrated with two different gas mixtures (room air (FiO_2_: 20.85%; FiCO_2_: 0.10%) and standardized gas in a different high-pressure bottle (FiO_2_: 16.00%; FiCO_2_: 4.00%)). Arterial oxygen saturation (SpO_2_) was continuously measured via a finger pulse oximeter (SenSmart 8100S Serie; Nonin, MN, USA). Heart rate (HR) was obtained with a 12-lead electrocardiogram (ECG) (Strässle & Co. DT 100, Albstadt, Germany), and blood pressure (BP) (Medisoft Ergoline 4M, Dinant, Belgium) was measured at the end of each stage.

The aim was to obtain a maximal test, and different criteria of hemodynamic, respiratory, metabolic or clinical limitations were considered. Limiting factors included ischemic electrocardiogram abnormalities, severe rhythm or conduction changes, or inadequate blood pressure (BP) response (systolic BP > 250 mmHg or systolic BP reduction during exercise for the two following steps), as well as clinical criteria such as pallor or profuse sweating. Metabolic limitation was defined as RER > 1.15 with an oxygen respiratory equivalent >40 or a VO_2_ plateau (an increase of less than 100 mL/min while workload increased further). Respiratory limitation was expressed as partial or complete consumption of the ventilatory reserve (VR) (VE max at the end of effort/maximal voluntary ventilation (MVV) > 70%), as well as desaturation indicated by an SpO_2_ < 75% at the end of the effort. Finally, muscular fatigue in the legs or respiratory discomfort indicated that the test was maximal but did not constitute a reason to stop. CPET was conducted to exhaustion if no criteria for stopping were present following the European and American guidelines [8,11].

### 2.4. Data Collection

The data collected for each patient were sociodemographic, including age and gender, body mass index (BMI), CPET indication, left ventricular ejection fraction (LVEF), and CPET parameters (VO_2_, VCO_2_,VE, and RER on a breath-by-breath file extracted from the device). VE/VCO_2_ slope calculations were determined using linear regression analysis of VE and VCO_2_ obtained up to VT1 (VT1-slope), VT2 (VT2-slope), and over the entire test (full-slope). To determine VT1 (i.e., aerobic threshold), the V-slope method established by Beaver et al. was used [12]. This consists of plotting the point cloud of VCO_2_ versus VO_2_ to detect the break that reflects a sudden increase in VCO_2_. VT2 (i.e., ventilatory compensation point) was determined via the Wasserman method using respiratory CO_2_ equivalents [12]; a sudden rise in the VE/VCO_2_ ratio, marked by a break in the curve, indicated VT2. Ventilatory thresholds were reviewed by two independent blinded exercise physiologists and the VO_2_ peak was expressed as % of predicted value (PV) according to the literature [13]. Patients were divided according to whether they surpassed VT2 (VT2 group) or not (VT1 group). Within both groups, we performed sub-analyses according to age and CPET indication (i.e., cardiac disease, lung disease or others, and dyspnea work-up).

### 2.5. Data Analysis

The Kolmogorov–Smirnov test confirmed that the continuous data distribution was non-gaussian. These data are expressed as the median and interquartile range [P25–P75]. We used the Mann–Whitney test to compare variables between patients who achieved VT2 (VT2 group) and those who did not (VT1 group). The Wilcoxon signed-rank test was used to compare variables within the two groups. The Spearman nonparametric correlation coefficient was used for correlation analysis. All analyses were performed using SPSS software version 22 (Chicago, IL, USA). A *p*-value < 0.05 was considered significant.

## 3. Results

### 3.1. General Characteristics of the Study Population

Among the 697 CPETs analyzed in this study, 389 patients did not reach VT2 (VT1 group) and 308 surpassed VT2 (VT2 group). Patients’ characteristics for each group are presented in Table 1. Patients of the VT1 group were older, had a higher BMI, drew less on their ventilatory reserve (VR), and achieved a lower VO_2_peak and RERpeak. Regarding the VE/VCO_2_ slopes, the VT1-slope was higher in the VT1 group than in the VT2 group. In contrast, there was no difference in the full-slope between the two groups.

### 3.2. Association between VE/VCO_2_ Slope and Age

Table 2 shows the VE/VCO_2_ slope values according to age. In the VT1 group, an increase in the VT1-slope and full-slope with age was observed. In the VT2 group, the same relationship between age and increases in VE/VCO_2_ slopes was observed. Considering all patients, we found a positive correlation between age and the VT1-slope (Spearman’s *ρ* = 0.201, *n* = 697, *p* < 0.0001), the VT2-slope (Spearman’s *ρ* = 0.191, *n* = 308, *p* < 0.001), and the full-slope (Spearman’s *ρ* = 0.137, *n* = 697, *p* < 0.0001).

### 3.3. Patient Characteristics According to Stress Test Indications

Table 3 shows the patient characteristics according to their indications for the stress test. Regardless of the stress-test indication, patients of the VT1 group were older, had a higher BMI, and reached a lower RERpeak and VO_2_peak. For VE, except for lung disease patients, the VT2 group drew more vigorously on their VR than the VT1 group. Regardless of the stress-test indication, the VT1-slope was higher in the VT1 than in the VT2 group. In contrast, there was no difference in the full-slope between the two groups.

### 3.4. Comparison of Slopes According to Stress Test Indications

For each stress-test indication in the VT1 group, the full-slope was higher than the VT1-slope (Table 4). Considering all patients in the VT1 group, the median difference between the VE/VCO_2_ full-slope and the VT1-slope was + 4 [2–6] units (*p* < 0.001).

For each stress-test indication in the VT2 group, the full-slope was higher than the VT1-slope and VT2-slope, and the VT2-slope was higher than the VT1-slope (Table 4). Considering all patients in the VT2 group, the VE/VCO_2_ full-slope was higher than the VT1-slope (+ 7 [5–11] units; *p* < 0.001) and the VT2-slope (+ 4 [2–5] units; *p* < 0.0001). In this group, the VT2-slope was higher than the VT1-slope (+ 3 [2–5] units; *p* < 0.0001).

### 3.5. Relationship between VO_2_ and VE/VCO_2_ Slope

Figure 1 shows a negative correlation between the full-slope and VO_2_peak in the whole population (Spearman’s ρ = 0.37, *n* = 697, *p* < 0.0001). This negative correlation was observed irrespective of whether patients achieved VT2 or not (Table 5).

## 4. Discussion

This cross-sectional study was carried out to assess and compare different methods of measuring the VE/VCO_2_ slope at the following time intervals: from rest (1) to VT1 (VT1-slope); (2) to VT2 when achieved (VT2-slope); and (3) over the entire test (full-slope). Regardless of the CPET test indication, the gradient of the VE/VCO_2_ slope was greater for all time intervals in both patients who surpassed VT2 as well as those who did not. Importantly, while the VT1 group presented a lower VO_2_ peak than the VT2 group, there was no difference regarding the full-slope in either group. As expected, there was a positive correlation between age and the VE/VCO_2_ slopes [14]. Finally, a negative correlation between the VE/VCO_2_ slope and VO_2_peak was also observed.

The steepness of the increase in VE as regards VCO_2_ (VE/VCO_2_ slope) is indicative of ventilatory efficiency and can identify an abnormal ventilatory response to exercise or increased physiological dead space. Measurement of the VE/VCO_2_ slope is a noninvasive, reproducible, prognostic marker in several chronic cardiopulmonary diseases, along with other exercise-related variables such as VO_2_peak [11]. We found a highly significant negative correlation between all the VE/VCO_2_ slopes and the VO_2_peak. This is in line with other studies supporting that the VE/VCO_2_ slope is an extremely useful prognostic tool in the VO_2_peak grey area (i.e., between 10 and 18 mL/min/kg) [15]. It is also thought that the VE/VCO_2_ slope calculated from data acquired below the ventilatory compensation point (i.e., VT2), determined from submaximal exercise, carries the same prognostic information as the VE/VCO_2_ slope calculated from a maximal test. Clinicians have been interested in obtaining the same prognostic information from submaximal exercise as currently obtained from maximal exercise to make testing more applicable to activities of daily living for patients who are unable or unwilling to perform a maximal exercise test [16].

Based on the alveolar ventilation equation, there are three variables that determine VE/VCO_2_: the physiological dead space ventilation relative to tidal volume (VD/VT), the amount of CO_2_ produced, and the arterial PaCO_2_. Many cardiopulmonary diseases present an altered PaCO_2_ set-point and chemosensitivity, which then influence VE/VCO_2_. Inefficient gas exchange caused by increased ventilation–perfusion heterogeneity increases VD/VT, thus contributing to the abnormal VE/VCO_2_ slope seen in cardiopulmonary diseases. Both the PaCO_2_ and VD/VT variables evolve continuously throughout exercise and are largely dependent on the level of effort achieved, which directly affects the value of the VE/VCO_2_ slope [17]. Our results show an increase in the VE/VCO_2_ slope between VT1 and VT2, which supports the fact that maximal and submaximal VE/VCO_2_ slopes are not equivalent and it is important to evaluate correctly the VT2 and VT2 points.

However, in many cardiorespiratory disease patients, maximal metabolic exercise testing is not feasible, and a final respiratory exchange ratio of <1.10 is common [7]. All our CPETs were performed with the objective of being maximal; nevertheless, more than half of our patients did not reach VT2. We still found that these patients had a median RERpeak of 1.16, confirming that our CPETs showed a consequent level of effort [18]. To compare the different VE/VCO_2_ slope measurement methods, we divided our sample into two groups depending on whether they had surpassed VT2 or not. Regardless of the CPET test indication, the gradient of the VE/VCO_2_ slope was greater for all time intervals in both patients who surpassed VT2 and those who did not. The increase observed between the VT1-slope and VT2-slope suggests that the relationship between VE and VCO_2_ is not fully linear up to VT2 [17]. While the mechanisms underlying this observation are not completely understood, the causes of the increase in the VE/VCO_2_ slope before and after VT2 are well characterized. VE is driven by acidosis and CO_2_ after VT2, signifying that the VE/VCO_2_ slope becomes much steeper (greater VE increase per unit of VCO_2_) [17]. Therefore, using the entire CPET (i.e., full-slope) to calculate the VE/VCO_2_ slope will produce a steeper slope than by using just the data from rest to VT1 or VT2 [16,17]. In both patient groups (VT1 and VT2), the mean difference between submaximal slopes (i.e., the VT1-slope and VT2-slope) and the full-slope was about 4 units, which is sufficient to reclassify a patient from a low-risk into a high-risk category for HF or pulmonary hypertension [19].

No standard definition has been established as to how the VE/VCO_2_ slope should be measured. Some researchers do not include data after identifying VT2 when reached. Here, subjectivity is introduced as data are only omitted from tests that identify VT2; thus, slopes could be calculated differently depending on the patient. In our evaluation of CPETs, the VE/VCO_2_ slope is a more significant prognostic tool; therefore, how it is measured must be standardized. The American Heart Association (AHA) stated that: “… calculation of the VE/VCO_2_ slope with all exercise data obtained from a progressive exercise test (initiation to peak effort) appears to provide additional clinical information compared with submaximal calculations (i.e., those that use linear data points before the steepening associated with ventilatory compensation for metabolic acidosis)” [20].

Like others, we found a strong correlation between the full-slope and submaximal slopes (VT1 vs. full slope (R = 0.84), VT2 vs. full slope (0.92) and VT1 vs. VT2 (0.88); *p* value < 0.001 for the three values). As the full-slope calculation considers all points used to calculate the submaximal slope, this correlation was expected. However, an important part of the “prognostic signal” appears to arise after VT2. It is not fully clear why; it might be because higher exercise levels in patients with poorer cardiac function produce greater acidosis with a steeper VE/VCO_2_ slope after VT2. Ingle et al. suggested that a significant predictor of outcome derives from the difference between the submaximal VE/VCO_2_ slope and the VE/VCO_2_ slope after VT2; the greater this difference, the worse the prognosis is. They concluded that all data points (i.e., full-slope) should be used to measure the VE/VCO_2_ slope and thus obtain its highest prognostic value [21]. When calculated over the entire test, we found that the VE/VCO_2_ slope (i.e., full-slope) was identical between patients who surpassed VT2 and those who did not. This is particularly relevant since our data show that less than half of patients achieved VT2 despite metabolically maximal exercise. This suggests that, even without reaching VT2, the full-slope is likely better fitted than the submaximal slope to assess prognosis in cardiopulmonary disease patients.

In patients not achieving VT2, the VE/VCO_2_ slope data acquired below VT2 provide a prognostic substitute; however, some prognostic sensitivity is lost. Not being able to reach VT2 yet having a high VE/VCO_2_ slope could be related to advanced illness and poor prognosis [21,22]. Patients who did not achieve VT2, compared with those who did, had a higher BMI, were on average 4.5 years older, and had a lower VO_2_peak of 12% and a lower RERpeak. This lower VO_2_peak cannot be explained just by their older age as a decrease in VO_2_max of 0.5 to 1% per year is generally admitted [23]. As suggested by the lower RERpeak obtained, the difference in VO_2_peak between the two groups could be partly explained by a less metabolically intense effort [2,11]. These patients were also probably unable to exercise to VT2 because of advanced illness, as suggested by a higher VT1-slope, likely associated with metabolic deconditioning. Like others, we observed a slight increase in the VE/VCO_2_ slope values with age, regardless of the measurement method and the test indication. This supports the use of relative VE/VCO_2_ slope values rather than absolute values [14]. A VE/VCO_2_ slope measured at 35 may be normal at 70 years but abnormal at 20 years. Increased pulmonary artery pressure, reduced cardiac output, increased dead space/tidal volume ratio, hypersensitivity to chemoreceptor activation, deconditioning, and muscle weakness are some of the many processes involved in the mechanisms underlying the VE/VCO_2_ slope increase with age [17,24].

### Strengths and Limitations

To our knowledge, this is the first study to assess the differences in VE/VCO_2_ slopes up to VT1, up to VT2, and over the entire test in different cardiopulmonary diseases. Our results clearly show that the VE/VCO_2_ slope measurement method used changes its value and should be homogenized to improve its prognostic power.

There are, however, several limitations of the present study that could affect the results or conclusions. First, this is a single-center retrospective study with a relatively small sample size; however, it offers the advantage of all parameters being measured with the same device and the same operators. Second, the VT1 and VT2 could be sometimes quite subjective. The thresholds have been reviewed by two independent physiologists to reduce the subjectivity part of the threshold’s determination. The two physiologists have determined threshold independently. When the thresholds were not located at the same level of workload, the two physiologists reviewed the thresholds together to agree on the same level of workload. Third, not all CPET centers use the same ramp protocol as studied herein [11]; therefore, the use of other testing modalities or protocols may affect the generalizability of our findings. Another issue involves calculating the VE/VCO_2_ slope via linear regression, which likely fits to a submaximal VE/VCO_2_ slope, yet the VE/VCO_2_ slope from an entire CPET is not linear [25,26]. However, the recommendations are to use the angular coefficient of the VE vs. VCO_2_ relationship and consider it as a linear regression. Therefore, there are no references for a nonlinear equation.

## 5. Conclusions

In conclusion, our data demonstrate the existence of great differences depending on the method used to calculate VE/VCO_2_ slopes. The significant variations observed certainly limit the prognostic value of the VE/VCO_2_ slope and homogenization of the calculation method seems critical to improve its prognostic value. The difference we observed between the full-slope and submaximal slopes was sufficient to reclassify HF patients from low-risk to high-risk categories. This supports the use of the data from the entire test (full-slope) to calculate the VE/VCO_2_ slope.

## Figures and Tables

**Figure 1 healthcare-11-01292-f001:**
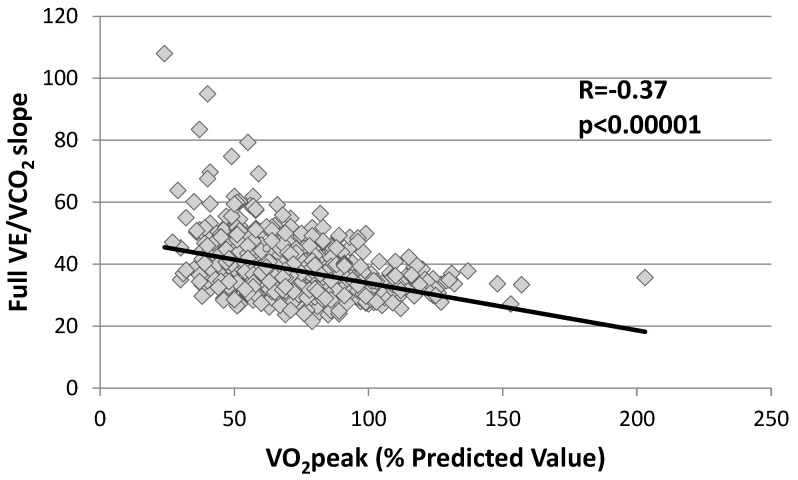
Correlation between VO_2_peak (% PV) and full VE/VCO_2_ slope in all patients (grey square). VE/VCO2 slopes was significantly correlated to VO2max. The black line represents the linear correlation curve.

**Table 1 healthcare-11-01292-t001:** General characteristics of the study population.

	Population (*n* = 697)	Group VT1 VT2 Not Reached (*n* = 389)	Group VT2 VT2 Surpassed (*n* = 308)	*p*-Value
Age (years)	57 [47–66]	59 [49–67]	54 [44–63]	<0.001
BMI (kg/m^2^)	27 [23–30]	27 [24–31]	26 [23–30]	<0.006
VO_2_ peak (% PV)	69 [57–83]	64 [53–78]	76 [64–88]	<0.001
RER peak	1.19 [1.13–1.26]	1.16 [1.1–1.2]	1.20 [1.17–1.28]	<0.001
VT1-slope	31 [27–35]	32 [29–37]	29 [25–33]	<0.001
VT2-slope	-	-	33 [29–37]	-
Full-slope	37 [33–42]	36 [33–42]	37 [33–41]	<0.870
VR (% MVV)	59 [49–72]	55 [47–67]	64 [54–74]	<0.001
Sex ratio M/F	1.27	1.08	1.56	0.031
LVEF (%)	60 [60, 60]	60 [58–60]	60 [60, 60]	0.55

**Table 2 healthcare-11-01292-t002:** Median VE/VCO_2_ slope values according to age for patients in the VT1 and VT2 groups.

	VT1 Group, *n* = 389			VT2 Group, *n* = 308			
Age (Years)	VT1-Slope	Full-Slope	% Difference VT1 vs. Full	VT1-Slope	VT2-Slope	Full-Slope	% Difference VT1 vs. Full/VT2 vs. Full
<30	31 [28–34]	34 [32–40]	9.67%	28 [24–34]	31 [27–38]	34 [31–42]	21.43/9.68
31–40	32 [29–35]	36 [31–40]	12.50%	28 [25–32]	31 [27–36]	35 [32–40]	25.00/12.90
41–50	30 [26–36]	36 [31–40]	20.00%	28 [25–31]	32 [29–34]	37 [33–41]	32.14/15.63
51–60	32 [30–36]	37 [33–41]	15.63%	28 [25–33]	32 [29–37]	36 [33–41]	28.57/12.50
61–70	33 [30–37]	37 [33–43]	12.12%	29 [27–36]	34 [30–39]	38 [35–43]	31.03/11.76
>70	34 [29–42]	38 [33–47]	11.76%	30 [26–34]	35 [30–38]	38 [34–41]	26.67/8.57

**Table 3 healthcare-11-01292-t003:** Patient characteristics according to stress-test indications.

	**Cardiac Diseases**			**Lung Diseases**		
	**VT1 Group** **(*n* = 191)**	**VT2 Group** **(*n* = 186)**	***p*-Value**	**VT1 Group** **(*n* = 70)**	**VT2 Group** **(*n* = 32)**	***p*-Value**
Age (years)	63 [51–69]	57 [47–66]	<0.001	62 [53–69]	55 [41–63]	0.006
BMI (kg/m^2^)	27 [24–31]	26 [23–29]	<0.001	25 [21–30]	26 [21–29]	0.003
VO_2_peak (% PV)	60 [51–76]	75 [64–89]	<0.001	65 [52–79]	74 [53–80]	0.034
RERpeak	1.17 [1.12–1.23]	1.22 [1.16–1.28]	<0.001	1.16 [1.1–1.21]	1.22 [1.2–1.26]	<0.001
VR (% VMM)	54 [46–65]	64 [54–74]	<0.001	59 [48–73]	62 [53–71]	0.785
VT1-slope	33 [29–38]	29 [26–37]	<0.001	34 [30–37]	28 [25–35]	0.04
VT2-slope	-	33 [30–34]	-	-	32 [29–40]	
Full-slope	37 [33–43]	37 [33–42]	0.899	38 [33–44]	37 [33–46]	0.562
	**Others**			**Dyspnea Work-Up**		
	**VT1 Group** **(*n* = 81)**	**VT2 Group** **(*n* = 64)**	***p*-Value**	**VT1 Group** **(*n* = 47)**	**VT2 Group** **(*n* = 26)**	***p*-Value**
Age (years)	53 [45–63]	52 [44–60]	<0.001	53 [46–64]	43 [34–52]	<0.001
BMI (Kg/m^2^)	29 [24–33]	27 [25–31]	0.003	27 [24–34]	26 [21–31]	0.035
VO_2_peak (% VP)	68 [57–83]	78 [64–88]	0.005	60 [56–80]	76 [66–100]	0.022
RERpeak	1.16 [1.09–1.21]	1.23 [1.17–1.33]	<0.001	1.15 [1.07–1.23]	1.23 [1.17–1.28]	0.001
VR (% VMM)	55 [48–66]	67 [57–79]	0.01	56 [49–67]	63 [52–76]	0.02
VT1-slope	31 [27–36]	27 [25–31]	0.014	31 [28–35]	29 [24–32]	0.014
VT2-slope	-	31 [29–35]	-	-	32 [28–35]	
Full-slope	36 [31–40]	36 [32–40]	0.105	36 [32–40]	36 [31–39]	0.347

**Table 4 healthcare-11-01292-t004:** Comparison of VE/VCO_2_ slopes according to test indications in VT1 and VT2 groups.

	VT1 Group, *n* = 389			VT2 Group, *n* = 308			
	VT1-Slope	Full-Slope	VT1 vs. Full	VT1-Slope	VT2-Slope	Full-Slope	VT1 vs. VT2/ VT2 vs. Full
Lung	34 [30–37]	38 [33–45]	<0.001	28 [25–31]	31 [29–37]	36 [32–41]	<0.001/<0.001
Others	31 [27–36]	36 [31–40]	<0.001	27 [25–31]	31 [29–35]	36 [32–40]	<0.001/<0.001
Dyspnea work up	31 [28–35]	36 [32–40]	<0.001	29 [24–32]	32 [28–35]	36 [31–39]	<0.001/<0.001
Cardiac	33 [29–38]	37 [33–47]	<0.001	29 [26–34]	33 [30–37]	38 [33–42]	<0.001/<0.001
All	32 [29–37]	36 [33–42]	<0.001	29 [25–33]	33 [29–37]	37 [33–41]	<0.001/<0.001

**Table 5 healthcare-11-01292-t005:** Correlations between VO_2_peak (% Predicted Value) and VT1-VT2 and full VE/VCO_2_ slope according to the level of effort achieved.

		VO_2_peak(% PV)		
		*n*	*r*	*p*-Value
All	VT1-slope	697	−0.41	<0.001
	full-slope	697	−0.37	<0.001
VT1 group	VT1-slope	389	−0.4	<0.001
	full-slope	389	−0.42	<0.001
VT2 group	VT1-slope	308	−0.32	<0.001
	VT2-slope	308	−0.32	<0.001
	full- slope	308	−0.33	<0.001

## Data Availability

Data are available upon demand from the authors.

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
