# Peer review of "How Does the Method Used to Measure the VE/VCO_2_ Slope Affect Its Value? A Cross-Sectional and Retrospective Cohort Study"

_healthcare, 2023, doi:10.3390/healthcare11091292_

Round 1
Reviewer 1 Report
It is a very well performed study with a fair amount of thought put into it. However, the objective of the study itself is not clinically significant with regards to adding to the abundant literature available for cardiopulmonary exercise testing. I do not feel the study warrants publication in this journal.
The study is very well conceptualized and well conducted retrospective analysis.
However, the reason for the rejection is a critical flaw in the objective of the study. The authors are looking to assess different ways and variables in the measurement of the Ve/VCO2 slope in a sub-maximal exercise study.
There is a notion that using the slope even in a submaximal exercise study is just as accurate as in a maximal exercise study. However, multiple studies have kind of debunked this theory.
More importantly, the AHA statement recommends that we should be calculating it using the data all the way up to peak exercise. Several studies indicate that although the start to AT (sub-maximal) Ve-VCO2 slope was a powerful predictor of an individual’s clinical outcome, the start to peak Ve-VCO2 slope was superior.
The reasons for this are speculated that it is due to poorer cardiac function at higher levels of exercise which leads to greater acidosis and a steeper Ve-VCO2 slope after anaerobic threshold. So the concept of trying to find ways to measure factors influencing the slope in this study which has a significant number of patients who did not reach anaerobic threshold is innately flawed and hence the rejection.
My recommendation would be to only add tests where subjects achieved maximal exercise.
Thank you
Author Response
Dear Reviewer,
We thank all the reviewers for their valuable comments and suggestions that have undoubtedly helped improve the manuscript.
Reviewer 1
It is a very well performed study with a fair amount of thought put into it. However, the objective of the study itself is not clinically significant with regards to adding to the abundant literature available for cardiopulmonary exercise testing. I do not feel the study warrants publication in this journal.
Response: In hospitals, healthcare must measure the VE/VCO2 slope to stratify mortality and hospitalization in different populations (HF, PH, COPD, thoracic surgery…). However, it is especially important, from our point of view, to consider the method used to calculate this slope and the impact of measuring the slope at different points. So, the study aimed to evaluate and compare different methods of measuring the VE/VCO2 slope, as mentioned in lines 69 to 79.
“To date, there are several methods of calculating the VE/VCO2 slope, varying from one study to another, and expert recommendations are not that clear. Some authors calculate the slope only using the data from the start of exercise to the aerobic threshold (i.e., first ventilatory threshold (VT1)) or the ventilatory compensation point (i.e., secondary ventilatory threshold (VT2)), while others use all the data from the start to the maximum effort (full-slope) [10]. Thus, during a single CPET, several VE/VCO2 slope measurements can be obtained depending on the method used. This measurement heterogeneity certainly limits the clinical relevance and prognostic value of the VE/VCO2 slope when the test is maximal. This work aims to evaluate and compare the different methods of measuring the VE/VCO2 slope, considering values from rest up to VT1 (VT1-slope), to VT2 (VT2-slope), and including all the data from the entire test (full-slope).
We also added a hypothesis, as recommended by reviewer 2. (Lines 79 to 82)
We formulated the primary hypothesis that the gradient of the VE/VCO2 slope is greater for all time intervals (i.e., up to VT1, VT2, and over the entire test). In addition, we speculated that the VE/VCO2 slope is influenced by age and cardiopulmonary disease.
The study is very well conceptualized and well conducted retrospective analysis.
Response: We thank the reviewer for their comments.
However, the reason for the rejection is a critical flaw in the objective of the study. The authors are looking to assess different ways and variables in the measurement of the Ve/VCO2 slope in a sub-maximal exercise study.
Response: The tests were not submaximal as reported by a mean maximal RER of 1.19 in the entire population, 1.16 in the VT1 group, and 1.20 in the VT2 group. Clinically, not all patients specifically reach VT2, or, in some cases, VT2 is not clearly definable. However, this does not mean that the test is submaximal (Guazzi, Circulation 2016). Moreover, we carried out the tests until exhaustion.
In our study, 308 patients (44%) achieved VT2. The remaining 389 patients either did not reach VT2 or VT2 was not clearly identifiable. In a previous book by Aguilaniu B, only about 20 to 30% of patients reach VT2, which is approximately the same as in our population.
There is a notion that using the slope even in a submaximal exercise study is just as accurate as in a maximal exercise study. However, multiple studies have kind of debunked this theory.
Response: We, however, do fully agree with this theory.
The VE/VCO2 slope, even in submaximal exercise, must be used as a primary parameter with VO2max in the analysis of CPET. In a paper by Guazzi (EACPR/AHA Scientific Statement, EHJ 2018), the VE/VCO2 slope was just as predictive of cardiovascular risk or death as VO2max for specific populations (HFPEF, HFMEF, HFREF) (Sarullo 2010, Arena 2007, Gong 2022).
Our paper shows that measurement of the VE/VCO2 slope can be different (> than 4 points and up to 8 points) when the slope is measured to VT1, VT2, or full data. As mentioned in the discussion, 4 points are sufficient to change the stratification category. Therefore, our paper contributes to debunk this theory and demonstrates it in different populations and ages.
More importantly, the AHA statement recommends that we should be calculating it using the data all the way up to peak exercise. Several studies indicate that although the start to AT (sub-maximal) Ve-VCO2 slope was a powerful predictor of an individual’s clinical outcome, the start to peak Ve-VCO2 slope was superior.
Response: The AHA statement 2010 cites: “Moreover, calculation of the V̇E/V̇CO2 slope with all exercise data obtained from a progressive exercise test (initiation to peak effort) appears to provide additional clinical information compared with submaximal calculations (i.e., those that use linear data points before the steepening associated with ventilatory compensation for metabolic acidosis). Although the former method affords optimal clinical information, the latter submaximal calculations have also demonstrated diagnostic/prognostic value.”
The ESC position statement cites: “The VE/VCO2 slope is most commonly calculated using all ET data”. And explains that “An inefficient VE typically translates into an abnormal rate of increase in the slope of VE increase versus VCO2 production. This relationship shows a near-linear increasing pattern that is determined by 3 factors: the amount of CO2 produced; the physiological dead space/tidal volume ratio (VD/VT); and the arterial carbon dioxide partial pressure (PaCO2). This relationship can be explained using the modified alveolar equation: VE= [863 x VCO2]/[PaCO2(1 - VD/VT)].
So, the explanations and propositions of two different scientific reports are not that clear. However, prediction scores are only based on the full VEVCO2 slope rather than the VEVCO2 slope calculated up to VT1 or VT2. Moreover, many studies calculate the VEVCO2 slope up to VT2 by linear regression.
The reasons for this are speculated that it is due to poorer cardiac function at higher levels of exercise which leads to greater acidosis and a steeper Ve-VCO2 slope after anaerobic threshold. So the concept of trying to find ways to measure factors influencing the slope in this study which has a significant number of patients who did not reach anaerobic threshold is innately flawed and hence the rejection.
My recommendation would be to only add tests where subjects achieved maximal exercise.
Response: We appreciate your observation, but this is exactly what we did as reflected by maximal metabolic tests ( RER > 1.10 in all tests) and maximal clinical tests because tests were performed to exhaustion, as recommended by the AHA scientific report. One of the exclusion criteria is an RER <1.10.
We thank the Reviewer for their time and constructive comments that have greatly improved our manuscript, which we hope will now be suitable for publication in Healthcare;
For more details please see the revised version manuscript.
Reviewer 2 Report
Manuscript title
How does the method used to measure the VE/VCO2 slope affect its value? A cohort study on 700 successive cardiopulmonary exercise tests
Manuscript ID: healthcare-2318255
Summary
This study aimed to determine whether different methods of determining VE/VCO2 slope affect its prognostic value. The study showed that the method used to determine VE/VCO2 slope had great influences in the results, limiting its prognostic value. Additionally, the results demonstrated that mean difference between submaximal and full-slopes was enough to reclassify people from low to high risk for heart failure and pulmonary hypertension. The authors concluded that calculation methods should be standardized to enhance prognostic capability
General comments:
The manuscript will benefit from an English revision and consistency in delivering information. There are many confusing sentences, compromising the reading flow. My main concern is that the authors used a liner regression for a non-linear response (VE/VCO2 full-slope). The authors need be more convincing about the relevance and importance of their results.
Title
My suggestion for the title. “How does the method used to measure the VE/VCO2 slope affect its value? A cross-sectional and retrospective cohort study.
Abstract
The abstract included objectives, methodology, key findings, and conclusion. However, the authors should introduce all the results and then the conclusion. For example, the authors wrote that “We conclude that the method used for determining VE/VCO2 slope greatly influences the result, the significant variations limiting itsprognostic value. The calculation method must be standardized to improve its prognostic value. The mean difference between submaximal and full-slopes was ±4 units, sufficient to reclassify patients from low to high-risk for heart failure or pulmonary hypertension.”
I suggest the authors to change the order (see below)
[...]. The mean difference between submaximal and full-slopes was ±4 units, sufficient to reclassify patients from low to high-risk for heart failure or pulmonary hypertension. We conclude that the method used for determining VE/VCO2 slope greatly influences the result, the significant variations limiting its prognostic value. The calculation method must be standardized to improve its prognostic value.
Introduction
The introduction is clear, stating the topic of the research and problem, and objectives. However, the novelty of the study and literature gaps need to be further explored. What is the novelty of this study? What does your research add to the literature? Please explore this information in the introduction further. Hypotheses are missing in the introduction. Please include your hypothesis.
It would be helpful to provide more context about why CPET parameters (e.g., VE/VCO2) is important, and how it is used in the clinical practice. Provide examples of studies determining VE/VCO2 and their main findings.
Lines 35-38. This sentence is long compromising the reading flow. Please rewrite this sentence with shorter sentences.
Methods
Did you use a retrospective study guideline? Please include this information and follow the guideline.
How was the metabolic cart system calibrated? Please describe flow and gas calibrations.
Please include the criteria used to determine inadequate blood pressure response during CPET.
Please can you clarify how was SpO2 measured?
Data collection. Lines 98. CPET parameters. Please be specifc.
Results
It will be interesting to calculate and report the percentage difference between VT1-slope and full-slope (table 2) and VT1-slope, VT2-slope, and full-slope (Table 3).
Please align the table content to facilitate analysis.
Table 4. VO2max or VO2peak?
Line 157. Table 6. VT-slope or VT1-slope?
Table 6. Peak slope or full-slope?
Include the correlation value in the figure.
Why did you just include correlations between VO2peak and full-slope? Why not do the same with VT1 and VT-2 slopes?
Discussion
Since VE/VCO2 slope is the focus of this manuscript, the authors should discuss in more depth what is known about it and what are the main issues associated with ventilatory threshold determination.
Line 168. Cross-sectional study should be also included in the methods, in the study setting session.
Lines 229-230. Are the study correlations strong? What is considered a strong correlation?
Lines 272-274. My concern is that the authors used a liner regression for a non-linear response (VE/VCO2 full-slope). How this linear regression approach would impact their results and interpretations?
Conclusion
Conclusion is aligned with the objective of the study.
References
The references formatting must be confirmed, in accordance with the instructions to authors for the journal.
Author Response
Dear Reviewer,
Reviewer 2
Summary
This study aimed to determine whether different methods of determining VE/VCO2 slope affect its prognostic value. The study showed that the method used to determine VE/VCO2 slope had great influences in the results, limiting its prognostic value. Additionally, the results demonstrated that mean difference between submaximal and full-slopes was enough to reclassify people from low to high risk for heart failure and pulmonary hypertension. The authors concluded that calculation methods should be standardized to enhance prognostic capability
Response: We thank the Reviewer for these positive comments about our study.
General comments:
The manuscript will benefit from an English revision and consistency in delivering information. There are many confusing sentences, compromising the reading flow. My main concern is that the authors used a liner regression for a non-linear response (VE/VCO2 full-slope). The authors need be more convincing about the relevance and importance of their results.
Response: We thank the reviewer for their incisive comment. We have adapted the text to remove any confusing sentences and hope that it now flows better.
Of course, we agree that the VEVCO2 slope is not fully linear for reasons that we explain in the discussion part of our article. However, the Experts’ position or recommendations (AHA and ESC) consider this relationship a linear regression and recommend clinicians to calculate it as a linear regression. Thus, we followed the practical recommendations to calculate our different VE/VCO2 slopes.
Title
My suggestion for the title. “How does the method used to measure the VE/VCO2 slope affect its value? A cross-sectional and retrospective cohort study.
Response: Thank you, we have now adapted the title to this recommendation.
Abstract
The abstract included objectives, methodology, key findings, and conclusion. However, the authors should introduce all the results and then the conclusion. For example, the authors wrote that “We conclude that the method used for determining VE/VCO2 slope greatly influences the result, the significant variations limiting itsprognostic value. The calculation method must be standardized to improve its prognostic value. The mean difference between submaximal and full-slopes was ±4 units, sufficient to reclassify patients from low to high-risk for heart failure or pulmonary hypertension.”
I suggest the authors to change the order (see below)
[...]. The mean difference between submaximal and full-slopes was ±4 units, sufficient to reclassify patients from low to high-risk for heart failure or pulmonary hypertension. We conclude that the method used for determining VE/VCO2 slope greatly influences the result, the significant variations limiting its prognostic value. The calculation method must be standardized to improve its prognostic value.
Response: Thank you, we have now modified the abstract accordingly, from lines 20 to 22.
Introduction
The introduction is clear, stating the topic of the research and problem, and objectives. However, the novelty of the study and literature gaps need to be further explored. What is the novelty of this study? What does your research add to the literature? Please explore this information in the introduction further. Hypotheses are missing in the introduction. Please include your hypothesis.
Response: Thank you for the opportunity to further explain our research. The novelty of our study lies in the fact that we calculated VEVCO2 slopes up to different levels in the CPET (VT1, VT2, or full data) during maximal exercise tests and compared slopes with each other and between different ages and diseases. To our knowledge, no other study has compared different VEVCO2 slopes in different categories of patients and at different ages. We have added our hypothesis to the introduction.
Lines 79 to 82: . We formulated the primary hypothesis that the gradient of the VE/VCO2 slope is greater for all time intervals (i.e., up to VT1, VT2, and over the entire test). In addition, we speculated that the VE/VCO2 slope is influenced by age and cardiopulmonary disease.
It would be helpful to provide more context about why CPET parameters (e.g., VE/VCO2) is important, and how it is used in the clinical practice. Provide examples of studies determining VE/VCO2 and their main findings
Response: Thank you for this constructive comment. We have added a paragraph about the VECO2 slope to explain its importance in clinical practice (lines 50 to 62)
“The relationship between VE and VCO2, expressed as the VE/VCO2 slope, has been identified and documented [9]. A normal ventilatory response of the VE/VCO2 slope should be under 30.0 [2]. Patients with cardiopulmonary disease often present an abnormal ventilation response, characterized by an increased VE/VCO2 slope; for any given VCO2 level, VE is greater than normal.
The VE/VCO2 slope, VO2peak, and presence of exercise oscillatory ventilation (EOV) are considered primary CPET variables, being strong predictors of adverse events in heart failure (Class 1 A) [2]. Also, the VE/VCO2 slope is responsive to pharmacological, surgical, and exercise interventions in HF patients and can be used to assess therapeutic efficacy (Class 1A) [2]. In patients with chronic HF, a Kaplan Meier analysis revealed a one-year cardiac-related mortality of 75% in patients with a VE/VCO2 slope >35.6 and 25% in those with a VE/VCO2 slope < 35.6; one-year cardiac-related hospitalization was 77% in patients with a VE/VCO2 slope > 32.5 and 23% in those < 32.5 [6].”
Lines 35-38. This sentence is long compromising the reading flow. Please rewrite this sentence with shorter sentences.
Response: We have modified the sentence to…
Early prognostic studies about CPET were limited to the analysis of maximal aerobic capacity by measuring peak oxygen consumption (VO2peak). This parameter is used to characterize the functional severity of heart failure (HF) and, also, as a criterion for listing heart transplantations and preoperative risk stratification.
Methods
Did you use a retrospective study guideline? Please include this information and follow the guideline.
Response: We used a Strobe Checklist that is available but not requested by the journal.
How was the metabolic cart system calibrated? Please describe flow and gas calibrations.
Response: We have added the calibration details to lines 108 to 112.
The flow in the pitot tube was calibrated with a 3.00 L syringe, which was annually calibrated and verified by the manufacturer. The O2 and CO2 fractions were measured by laser cells calibrated against two different gas mixtures (room air FiO2: 20.85%; FiCO2: 0.10%) and standardized gas in a different high-pressure bottle (FiO2: 16.00%; FiCO2: 4.00%)
Please include the criteria used to determine inadequate blood pressure response during CPET.
Response: We have clarified the inadequate BP response during CPET in lines 120 to 121.
...or inadequate blood pressure (BP) response (systolic BP >250 mmHg or diminution of BP during exercise for the two following steps)
Please can you clarify how was SpO2 measured?
Response: We have added information regarding heart rate, SpO2, and BP measurements to lines 112 to 116.
Arterial oxygen saturation (SpO2) was continuously measured via a finger pulse oximeter (SenSmart 8100S Serie; Nonin, MN, USA). Heart rate (HR) was obtained with a 12-lead electrocardiogram (ECG) (Strässle & Co. DT 100, Albstadt, Germany), and blood pressure (BP) (Medisoft Ergoline 4M, Dinant, Belgium) was measured at the end of each stage.
Data collection. Lines 98. CPET parameters. Please be specifc.
Response: We have clarified the CPET parameters in lines 133 to 134.
CPET parameters (VO2, VCO2, and VE on a breath-by-breath file extracted from the device). RER was calculated as the VCO2 / VO2 ratio.
Results
It will be interesting to calculate and report the percentage difference between VT1-slope and full-slope (table 2) and VT1-slope, VT2-slope, and full-slope (Table 3).
Response: Thank you for this suggestion. We kept the direct values to be closer to reality and the values known by the clinical physician, however, we have added a column with the % differences in Tables 2 and 4.
Please align the table content to facilitate analysis.
Table 4. VO2max or VO2peak?
Response: We adjusted for VO2peak in the entire text to be more fluid.
Line 157. Table 6. VT-slope or VT1-slope?
Response: We adjusted for VT1-slope.
Table 6. Peak slope or full-slope?
Response: Thank you for pointing this out. We have corrected for full-slope throughout the text.
Include the correlation value in the figure.
Response: We have added the coefficient of correlation (R=0.37) and the p-value to the figure (p<0.0001).
Why did you just include correlations between VO2peak and full-slope? Why not do the same with VT1 and VT-2 slopes?
Response: Figure 1 illustrates VO2peak (% Predicted Value) vs. Full VEVCO2 slope. We calculated the correlations in Table 5, however, for the clarity of the text, we decided to illustrate only one correlation.
Discussion
Since VE/VCO2 slope is the focus of this manuscript, the authors should discuss in more depth what is known about it and what are the main issues associated with ventilatory threshold determination.
Response: Thank you for this comment. During exercise, VE responds to the changing rate of CO2 delivery to the lungs, including that generated by aerobic oxidation of energy substrates and HCO3-buffering of lactic acidosis. In addition, the carotid bodies monitor arterial [H*], providing sufficient ventilatory drive to minimize the decrease in arterial pH that occurs above VT1. Exercise ventilation is also determined by the size of the physiological Death Volume and the level at which arterial PCO2, is regulated. Pao2, normally remains relatively constant during exercise, despite increasing VO2.
VT1 is reduced when cardiovascular O2 transport is impaired (in patients). In addition, importantly exercise beyond VT1 increases CO2 and H* production, both being powerful ventilatory stimuli (Wasserman, JAPP, 1973)
As we also explained in the discussion, the VEVCO2 slope is an important prognostic factor, even more so when VO2max is affected (in the grey zone: 11<X>18 ml/min/kg). However, the other point is to correctly determine VT1 and VT2 points after CPET. The slope calculations also depend on the threshold determination: they should be performed by an experienced physiologist to avoid under or over risk estimation by a respectively under or over stratification.
The AHA statement is a bit vague on this point in that it says that a peak Ve-VCO2 slope less than 30 is normal and one greater than 40 is abnormal, leaving the actual upper limit of normal somewhat up in the air. This is something that needs further research since I have seen several studies using peak Ve-VCO2 slope that used an ULN of 34?
Of course, we agree with your opinion, and this is an important point. We demonstrate that a cardiac patient (but it is quite similar in other categories) could have a slower slope if the patient is not motivated to exercise to VT1, or if the healthcare professional does not calculate the slope to VT2 or full-slope.
Line 168. Cross-sectional study should be also included in the methods, in the study setting session.
Response: We have adjusted this part in the methods, line 85.
Lines 229-230. Are the study correlations strong? What is considered a strong correlation?
Response: Of course, the reviewer is right. In this part of the discussion, we examined the correlation between slopes measured between VT1 vs. full-slope (R=0.84), VT2 vs. full-slope (0.92), and VT1 vs. VT2 (0.88). Therefore, there is an evident strong correlation because R was over 0.75. We have added the correlations to the text to justify the term “strong correlation”, lines 282 to 283.
Lines 272-274. My concern is that the authors used a liner regression for a non-linear response (VE/VCO2 full-slope). How this linear regression approach would impact their results and interpretations?
Response: Naturally, we agree with the reviewer’s comment. But as explained previously, the recommendations are to use the angular coefficient of the VE vs. VCO2 relationship and consider it a linear regression. So, there are no references for a nonlinear equation, and nonlinear equations are not used in the clinic.
Conclusion
Conclusion is aligned with the objective of the study.
Response: We thank the reviewer for their comments.
References
The references formatting must be confirmed, in accordance with the instructions to authors for the journal.
Response: We have checked the references and adapted them for the journal.
For more details please see the revised version manuscript.
Reviewer 3 Report
This is a study comparing the different methods of measuring the VE/VCO2 slope in CPET. The authors provide an overall well-written manuscript. The study uses a relatively large sample size of 697 CPETs and the statistical methods are sound. My only minor suggestion is that it would have been good to have more clarity in the methods section on what the indication for CPET was - the authors mention they included patients referred for CPET as part of their "medical workup." It would have been good to understand why they were undergoing that work-up. The figures and tables are clear and informative but also wonder if tables could be condensed down to fewer tables (currently 7).
Author Response
Dear Reviewer,
Reviewer 3
This is a study comparing the different methods of measuring the VE/VCO2 slope in CPET. The authors provide an overall well-written manuscript. The study uses a relatively large sample size of 697 CPETs and the statistical methods are sound. My only minor suggestion is that it would have been good to have more clarity in the methods section on what the indication for CPET was - the authors mention they included patients referred for CPET as part of their "medical workup." It would have been good to understand why they were undergoing that work-up. The figures and tables are clear and informative but also wonder if tables could be condensed down to fewer tables (currently 7).
Response: We thank the reviewer for their comments. We have adapted the methods section according to the reviewer’s comments.
We have clarified the “participants” section: ...cardiac patients were referred for an HF or pulmonary hypertension follow-up, or the beginning of cardiac rehabilitation after ischemic or valvular surgery; pulmonary populations were addressed for pulmonary fibrosis, chronic obstructive pulmonary disease (COPD) follow-up, or pre-operative thoracic surgery; the dyspnea group was composed of patients addressed for exercise testing after a normal cardiac and pulmonary screening: and the last group (others) was composed of oncology, obesity (but normal cardiac screening), or rheumatoid arthritis patients.
We have also clarified the “procedure” and “data collection” sections according to the reviewer.
Old Tables 2 and 3 and Tables 5 and 6 have been combined to reduce the number of tables.
For more details please see the revised version manuscript.
Round 2
Reviewer 2 Report
General comments:
The manuscript flow and information improved. The authors included most of the suggestions. There are still a few things that need minor revisions.
Introduction
Lines 35-36. The sentence says, “Early prognostic studies about CPET were limited to the analysis of maximal aerobic capacity by measuring peak oxygen consumption.” It is important to mention that CPET does not provide information about maximal aerobic capacity. It is a measurement of maximal or peak aerobic power. Please change your maximal aerobic capacity for maximal or peak aerobic power.
Line 36. Include a space after peak oxygen consumption(VO2peak).
Line 75. If you are measuring the VE/VCO2 slope considering age and cardiopulmonary disease, are you speculating or hypothesizing? I recommend you reflect and change accordingly.
Methods
You should include this information in the manuscript if you used the Strobe checklist.
Lines 102-103. The flow in the pitot tube was calibrated with 102 a 3.00 L syringe, which was annually calibrated and verified by the manufacturer. Please remove .00.
Line 114. Please change diminution of BP for BP reduction during exercise. How about diastolic blood pressure? You should include DBP termination criteria.
Lines 122-123. “CPET was conducted to exhaustion if no criteria for stopping, according to the guidelines, were presented [11],” I suggest CPET was conducted to exhaustion if no criteria for stopping were present following the xxxx guidelines. Which guidelines are you following? Please include this information.
Lines 127-128. “CPET parameters (VO2, VCO2 and VE on a breath-by-breath file extracted from the device). RER was calculated as VCO2, VO2 ratio”. I suggest CPET parameters (VO2, VCO2, VE, and RER)
Results
Please review Table 3 format.
Table 4. The table caption font size is different. Please correct it.
Discussion
Lines 258-259. Maybe I missed it, but this is the first time you have presented these correlations.
Lines 301. We tried to minimize this bias by two independent lectures of VT1 and VT2. How these independent lectures reduced the bias? Please explain.
Author Response
General comments:
The manuscript flow and information improved. The authors included most of the suggestions. There are still a few things that need minor revisions.
We thank the reviewer for their great comments who improve the quality of this work.
Introduction
Lines 35-36. The sentence says, “Early prognostic studies about CPET were limited to the analysis of maximal aerobic capacity by measuring peak oxygen consumption.” It is important to mention that CPET does not provide information about maximal aerobic capacity. It is a measurement of maximal or peak aerobic power. Please change your maximal aerobic capacity for maximal or peak aerobic power.
We modified the sentence accordingly.
Line 36. Include a space after peak oxygen consumption (VO2peak).
We corrected this fault.
Line 75. If you are measuring the VE/VCO2 slope considering age and cardiopulmonary disease, are you speculating or hypothesizing? I recommend you reflect and change accordingly.
We modified the sentence into hypothesized who appears to be more precise.
Methods
You should include this information in the manuscript if you used the Strobe checklist.
We added this information on line 82.
Lines 102-103. The flow in the pitot tube was calibrated with 102 a 3.00 L syringe, which was annually calibrated and verified by the manufacturer. Please remove .00.
We corrected the sentence.
Line 114. Please change diminution of BP for BP reduction during exercise. How about diastolic blood pressure? You should include DBP termination criteria.
We modified the sentence accordingly.
However, to our knowledge, the diastolic pressure is relatively stable or could maybe decrease during an aerobic exercise. But there is no strict criteria for an abnormal response of the diastolic pressure during exercise. In the literature, an increase of diastolic pressure is associated with hypercholesterolemia of latent ventricular dysfunction. Secondly, at the end of exercise, the diastolic pressure measured on the left arm could increase due to the movement of the left arm. So we didn’t use diastolic criteria to stop exercise.
Lines 122-123. “CPET was conducted to exhaustion if no criteria for stopping, according to the guidelines, were presented [11],” I suggest CPET was conducted to exhaustion if no criteria for stopping were present following the xxxx guidelines. Which guidelines are you following? Please include this information.
We added a reference and clarified the sentence.
Lines 127-128. “CPET parameters (VO2, VCO2 and VE on a breath-by-breath file extracted from the device). RER was calculated as VCO2, VO2 ratio”. I suggest CPET parameters (VO2, VCO2, VE, and RER)
We modified the sentence.
Results
Please review Table 3 format.
We reviewed the Table 3
Table 4. The table caption font size is different. Please correct it.
We corrected the table caption font size.
Discussion
Lines 258-259. Maybe I missed it, but this is the first time you have presented these correlations.
Yes, it was the first time we presented these correlations after the reviewer 2 comment about the strong correlation in the first-round revision. We highlighted our comments by the calculation of the VT1, VT2 and full slope correlations.
Lines 301. We tried to minimize this bias by two independent lectures of VT1 and VT2. How these independent lectures reduced the bias? Please explain.
The determination of the ventilatory thresholds can be different between several physiologists and this even when using common techniques (VT-slope or ventilatory equivalent). The thresholds have been reviewed by two independent physiologists to reduce the subjectivity part of the threshold’s determination. The two physiologists have determined threshold in independently. When the thresholds were not located at the same level of workload, the two physiologists reviewed the thresholds together to agree on the same level of workload. This method will decrease the potential mistake due to the experimenters.
We added this sentence in the limitation part of the study.